# The 10-year course of mental health, quality of life, and exile life functioning in traumatized refugees from treatment start

**Marianne Opaas[1]\*, Tore Wentzel-Larsen[1,2], Sverre Varvin[3]**

**1** Norwegian Centre for Violence and Traumatic Stress Studies (NKVTS), Oslo, Norway, **2** Centre for Child and Adolescent Mental Health, Eastern and Southern Norway, Oslo, Norway, **3** OsloMet–Oslo Metropolitan University, Oslo, Norway

\* marianne.opaas@nkvts.no

**Data Availability Statement:** All relevant data are within the manuscript and its Supporting Information files.

## Abstract

Refugee patients with severe traumatic experiences may need mental health treatment, but treatment results vary, and there is scarcity of studies demonstrating refugees' long-term health and well-being after treatment. In a 10-year naturalistic and longitudinal study, 54 multi-origin traumatized adult refugee patients, with a background of war and persecution, and with a mean stay in Norway of 10.5 years, were recruited as they entered psychological treatment in mental health specialist services. The participants were interviewed face-to-face with multiple methods at admittance, and at varying points in time during and after psychotherapy. The aim was to study the participants' trajectories of symptoms of post-traumatic stress, anxiety and depression, four aspects of quality of life, and two aspects of exile life functioning. Linear mixed effects analyses included all symptoms and quality of life measures obtained at different times and intervals for the participants. Changes in exile life functioning was investigated by exact McNemar tests. Participants responded to the quantitative assessments up to eight times. Length of therapy varied, with a mean of 61.3 sessions ($SD = 74.5$). The participants improved significantly in symptoms, quality of life, and exile life functioning. Improvement in symptoms of posttraumatic stress, anxiety, and depression yielded small effect sizes ($r = .05$ to $.13$), while improvement in quality of psychological and physical health yielded medium effect sizes ($r = .38$ and $.32$). Thus, long-time improvement after psychological therapy in these severely traumatized and mostly chronified refugee patients, was more notable in quality of life and exile life functioning than in symptom reduction. The results imply that major symptom reduction may not be attainable, and may not be the most important indication of long-term improvement among refugees with long-standing trauma-related suffering. Other indications of beneficial effects should be applied as well.

## Introduction

Research among refugees hosted in Europe, North-America, and Australia has shown an over-representation of mental health suffering among individuals with a refugee background

**Funding:** The author(s) received no specific funding for this work.

**Competing interests:** The authors have declared that no competing interests exist.

compared to the majority populations [1–3], even after many years in the host country [4]. This implies a challenge to the mental health system. In this paper, we examine the long-term trajectories of mental health, quality of life, and two aspects of functioning in exile of resettled refugees who sought treatment for trauma-related disorders in Norwegian mental health settings more than 10 years ago. At that time, there were only about 125 000 individuals with a refugee background in Norway, and the largest groups originated from Iraq, Somalia, Bosnia-Hercegovina, and Iran. Now the number is almost twice as high [5].

Refugees flee from conflict, persecution, violence, imprisonment, torture, and other human rights violations—sometimes from experiences beyond our imagination. Moreover, they often cope with severe dangers and obstacles, and mobilize substantial resources, strength, and adaptability on their way to refuge [6, 7]. Some are able to maintain their psychological equilibrium after adverse and dramatic experiences and may show considerable resilience [8, 9]. However, the multiple losses inherent in having to leave all that is known behind, adverse experiences during flight, and the challenges and hardships in exile add to the burden refugees need to cope with [10]. Together, this seriously challenges afflicted refugees' capacity for resilience. A sizeable proportion develop symptoms of posttraumatic stress, depression and other mental health disorders, and may be in need of mental health treatment, but still, most traumatized refugees go untreated [11].

In addition to war-related trauma preceding flight, many refugees have suffered adverse and potentially traumatic experiences during childhood and adolescence [12, 13]. Many refugees have thus been exposed to what is referred to as complex trauma [14], which includes exposure to multiple, cumulative, and prolonged human-caused potentially traumatic experiences such as childhood abuse, domestic violence, sexual and physical assault, persecution, torture, combat, and other life threatening experiences. Exposure to complex trauma may lead to a decreased threshold for PTSD, and problems of personality functioning, such as affect dysregulation, relational difficulties, identity disturbance, and problematic behaviour [15]. The ACE (Adverse Childhood Experiences) study has convincingly demonstrated that childhood adversities above a certain level predict psychological and physical health problems, reduced prosperity, and premature death [16].

Research has shown that a higher exposure to traumatic experiences before and after migration is associated with PTSD, depression and anxiety [1]. Up to one third of the refugee population may have posttraumatic stress disorder (PTSD), often comorbid with other anxiety-related disorders and depression [17]. Physical health disorders [18, 19] and physical pain [20] are also frequent among refugees. Berthold et al. [21] found that the number of physical health problems was doubled in refugees with PTSD and depression, compared to refugees without these mental health disorders. Furthermore, chronic pain was found to be frequent among multi-traumatized refugees in outpatient mental health clinics in Norway, and the level of pain was positively related to their symptomatology and severity of psychiatric morbidity [22]. The extent of mental and physical symptoms and problems of living in exile have severe consequences for the refugees' wellbeing and capacity to function [23, 24].

Studies including measures of quality of life (QOL) have shown that refugees' QOL is lower [25], and in clinical samples way below international population norms [12, 26]. Furthermore, a significant negative relationship between PTSD symptoms and subjective QOL has been demonstrated [27, 28]. The more severe the refugees' mental health symptoms, the poorer their quality of life [25]. Refugees' frequent psychosocial problems and exposure to post-migration stressors, such as unemployment, poverty, unstable housing, discrimination, social marginalisation, exclusion, and weak social network, impact negatively on their mental health, adaptation and well-being [6], and may diminish their ability to recover from pre-migration trauma [29]. Among multi-traumatized refugee patients, Teodorescu et al. [30] found that

unemployment, weak social network and weak social integration into the Norwegian community were significantly associated with more psychiatric morbidity and poorer quality of life.

Several studies have shown the interdependence between mastering the language of the host country, completing education or holding employment and mental health, well being, and functioning well in exile [31–33]. After resettlement, refugees are expected to attend programs to learn the exile language and then get financially independent, but refugees with mental health problems may have difficulties fulfilling these expectations. In a Swiss study, difficulties with communication and employment, together with isolation, were found to be among the most serious difficulties for traumatized refugee patients [31]. The authors proposed a circular association between mental disorders connected with posttraumatic and postmigratory stress, functional impairment, and postmigration living difficulties.

Reviews, meta-studies, and systematic reviews of mental health treatment outcome among refugees with trauma-related disorders, have shown positive mean effects on mental health symptoms [34–39], although within studies variation in individual participants' improvement is commonly great [40, 41]. What most reviews have in common is a compilation of studies with relative short follow-up after termination of the intervention (usually up to a few months) and with outcome measures restricted to mental health symptoms (and sometimes somatic symptoms). A few single clinical studies with longer follow-up, report improved outcome with time. A 10-year follow-up [42] of traumatized refugee patients evidenced a 30% decrease in participants qualifying for a PTSD diagnosis. The study demonstrated that improvement in this patient group may take time, and continue over time, but at the same time, it is noteworthy that 56% retained their diagnosis of PTSD and 69% their diagnosis of depression, evidencing significant chronicity. Correspondingly, Buhmann et al. [43] found no significant effect immediately following intervention, but the six and 18 months follow-ups of their study [44] demonstrated a small improvement over time in PTSD, depression and anxiety symptoms, and in level of functioning. However, the majority of the participants still had symptom scores above clinical cut-off. These studies indicate that many or most refugee patients retain their diagnosis/-es after treatment and only some improve clinically significantly–that is–to an amount that makes a difference to their lives. Therefore, longer follow-up is necessary to study improvement after therapy in this patient group.

The comorbidity of refugees' mental health problems and physical ailments, the daily life struggles in exile, and the complex needs of refugees and asylum seekers are acknowledged among refugee researchers [17, 38]. To reach clinically significant improvement, patients with complex disorders and difficult psychosocial living conditions need longer duration of therapy than what is ordinarily offered [45], and many may not improve to the level that is expected in other patient groups [46]. There is, however, no simple relationship between mental health improvement and therapeutic dose (e.g., number of sessions). Thus, studies on refugee patients investigating the effect of therapeutic dose on outcome showed only weak and inconsistent findings [47, 48]. Psychotherapy research in general [46] has demonstrated that after the first few sessions, other factors within and outside therapy begin to merge with sheer exposure to therapy in terms of number of sessions.

According to researchers in the refugee field, treatment approaches need special tailoring and additional approaches to be of benefit to the complex trauma and life situations of refugees [17, 29, 43]. The potential helpfulness of psychotherapy for traumatized refugees rests, at a basic level, then, on having enough therapy resources to be able to offer the necessary amount of sessions, having a reservoir of interventions that can be adjusted to individual needs, and on the therapist's skills in engaging the patient so that he or she remains in therapy until positive change is accomplished.

The short- and long-term effects of psychological treatment for refugees and what constitutes positive outcomes are only starting to be investigated [49]. Of note is also that patients'

evaluations of "good outcome" or "no change" may not coincide with therapists' evaluations or with symptoms changes [50]. Researchers in the field of refugee mental health have underlined the need to do more research into refugees' optimal health outcomes, the mechanisms of improvement and chronicity, the durability of treatment effects, and to not only focus on PTSD, but to study a wider scope of therapy outcome, including improvement in other aspects of mental health, functioning, relational security, and quality of psychological, physical, and social well-being [35, 51, 52]. More knowledge is thus needed about whether and how psychological therapies can be helpful, about outcomes in addition to symptoms improvement, such as psychological and physical wellbeing, relational and exile life functioning, and about the effect of treatment over time. The present study is an attempt to fill this knowledge gap. The aim was to examine the long-term changes in mental health, quality of life, and in two aspects of exile life functioning (exile language competence and employment/study status) among refugee patients in a naturalistic follow-up study from treatment start in mental health treatment units, through treatment, and for some time after termination.

## Method

In this study, individuals with backgrounds as asylum seekers or refugees will briefly be termed *refugees*. Moreover, the term 'traumatized' generically refers to having experienced severe, repeated, diverse, and potentially traumatizing events, preceding the development of mental health disorder.

### Design

In a longitudinal and naturalistic study from 2006 to 2017, 54 traumatized refugee patients responded to repeated interviews comprising qualitative and quantitative research methods assessing the course of their mental health, quality of life, and exile life functioning. Their therapists were also interviewed one or more times. The study was designed to obtain a comprehensive understanding of refugee patients' mental health and wellbeing over time, in context of their past and present life circumstances, their traumatic experiences, and their therapy experience. The treatment consisted of psychological therapy/psychotherapy and was conducted under naturalistic conditions in publicly funded, outpatient mental health services in the Oslo area, Norway. There are no specialized treatment units for traumatized refugees in Norway; treatment is provided within the general mental health services.

Participants were consecutively recruited into the study during the years 2006 to 2009. Inclusion criteria were: patients with severe mental health problems, age 18 to 65, with a history as refugee/asylum seeker with potentially traumatic experiences related to war, persecution, torture and other severe human rights violations, newly accepted for mental health treatment. A PTSD diagnosis was not required as the study sought to include various mental health problems that might ensue among refugees in the aftermath of potentially traumatic events. An ability to communicate in Norwegian was not required. Exclusion criteria were presence of psychosis or drug abuse, if severe enough to hinder research assessment.

The study was approved by the Norwegian South-East Regional Committee for Medical and Health Research Ethics [53] and therapy was conducted according to the Declaration of Helsinki: Ethical Principles for Medical Research Involving Human Subjects [54]. Participants' written, informed consent was required to take part in the study. The first consent was limited to three years. After three years, the participants were asked if they wanted to continue taking part in the study, and those who accepted, gave a new, non-time-limited informed and written consent.

Results from intake assessment [12, 55] and from 1- and 3-year follow-up [56] have previously been published. The pre-treatment analyses demonstrated that the participants had suffered extensive potentially traumatic events related to war and persecution [55] as well as adverse and potentially traumatic events during childhood [12]. The three-year follow-up study [56], including symptoms of posttraumatic stress disorder (PTSD), anxiety, and depression, and four aspects of quality of life (QOL) revealed significant group-level improvement in Depression, Anxiety, PTSD Arousal symptoms, and in the QOL domains Physical Health, Psychological Health, and Social Relationships. Length of treatment varied, and at the three-year follow-up, 29% of the therapies were still not terminated. For further details and a discussion of the findings, see [56, 57].

By 2017, all therapies were terminated, and most of them had been terminated for five or more years. The current study encompasses all quantitative follow-up data from the ten-year data collection period.

### Participants

The participants, 35 men and 19 women, came from 15 different countries in Asia, Europe, and Africa. Of these, 57.4% ($n = 31$) came from countries in the Middle East, 16.7% ($n = 9$) came from other Asian countries, 16.7% ($n = 9$) came from countries in the Balkans, and 9.3% ($n = 5$) came from various African countries. According to their responses to the Harvard Trauma Questionnaire checklist (HTQ, Part 1) [65] on traumatic events, the participants had experienced a mean of more than 16 different kinds of potentially traumatic events related to war, persecution and human rights violations, mostly in early adulthood, including exposure to torture in 52.8%, and rape in 20.8% of the participants. In addition, 92% of the participants had suffered various adverse and potentially traumatic experiences during their childhood, such as loss of parents, severe illness or accidents, domestic violence, and witnessing violence and killings outside the home [12]. At the time of intake to the study, mean age was 39.3 ($SD = 8.2$), mean stay in Norway was 10.5 years ($SD = 6.5$), 50.0% of the group could communicate in Norwegian, and 22.2% held employment, though on sick leave at treatment start. None was engaged in formal studies. One participant lived in a refugee reception centre. The others lived in ordinary flats in the community. Two persons did not hold a Norwegian citizenship or a non-time-limited permission to stay in Norway. Two of the participants had completed secondary education in Norway [12]. See Table 1 for demographics and scores on mental health, QOL, host country language competence and work/study status upon entering the study.

### Procedures

For the study, we collaborated with two outpatient clinics and six psychotherapists with publicly funded, independent practices, all part of the specialist mental health services in Norway. During the 3-year intake period in these services, all patients who were eligible according to the inclusion criteria, were asked to participate in the study. Potential participants were carefully informed, in writing and verbally, about the purpose of the study, about confidentiality, that participation in the study was voluntary, that rejecting was of no consequence for their treatment, and that, if they signed up for the study, they could withdraw at any time. Seventy-two patients were asked to participate in the study, and 54 accepted and gave their written consent (75% inclusion).

The participants were interviewed face-to-face throughout the study by two researchers who were also clinical specialists, and who were external to the services where the participants received their therapy. Information on the language ability of participants was provided by the

**Table 1. Demographics, mental health symptoms, quality of life, and indicators of functioning in exile upon entering therapy and the study.**

| Variables | N | M (SD) | Percent (n) |
|---|---|---|---|
| *Demographics*: | | | |
| Male gender | 54 | | 64.8 (35) |
| Age (years) | 54 | | 39.3 (8.2) |
| Married/Cohabitating | 54 | | 64.8 (35) |
| Participants with children | 54 | | 81.5 (44) |
| Years of education in country of origin | 54 | 9.7 (4.5) | |
| Years living in Norway | 54 | 10.5 (6.5) | |
| Experienced childhood trauma[a] | 54 | | 91.7% (49) |
| HTQ-Trauma Events[b] | 54 | 16.3 (6.3) | |
| *Symptoms*: | | | |
| PTSD-Total[c] | 52 | 2.82 (.47) | |
| PTSD-Reexperiencing[c] | 52 | 2.89 (.69) | |
| PTSD-Arousal[c] | 52 | 3.12 (.49) | |
| PTSD-Avoidance[c] | 52 | 2.56 (.57) | |
| Anxiety[d] | 53 | 2.89 (.59) | |
| Depression[d] | 53 | 2.94 (.54) | |
| *Diagnostic level of symptoms*: | | | |
| PTSD (clinical cutoff ≥2.5)[c] | 52 | | 78.9 (41) |
| Anxiety (clinical cutoff >1.75)[d] | 53 | | 96.2 (51) |
| Depression (clinical cutoff >1.75)[d] | 53 | | 98.1 (52) |
| *Quality of life* | | | |
| Physical Health[e] | 52 | 28.5 (13.8) | |
| Psychological Health[e] | 52 | 25.6 (15.9) | |
| Social Relationships[e] | 52 | 36.6 (23.4) | |
| Environment[e] | 52 | 45.2 (40.1) | |
| *Indicators of functioning in exile*: | | | |
| Communicated in Norwegian | 54 | | 50.0 (27) |
| Employed or studying | 54 | | 22.2 (12) |

[a]Percentage and number of participants who reported childhood adverse events [12].

[b]Harvard Trauma Questionnaire, Part 1 [58]: List of 37 types of experiences related to war, persecution and human rights violations, each confirmed experience is scored 1.

[c]HTQ-Part IV [58], 16 items, presence and severity of item/symptom scored 1 (*not at all*) to 4 (*extremely*)

[d]HSCL-25 [58], also scored 1 (*not at all*) to 4 (*extremely*).

[e]WHOQOL-Bref [59], 26 items, quality of life for each item scored 0 (*very poor*) to 100 (*excellent*).

mental health clinics, and interpreters were summoned accordingly. For participants who were expected to master the Norwegian language (or English), we arranged a first meeting without an interpreter, but if mutual understanding and expressive capacity in a common language was not sufficient, the meeting was rescheduled, and later research interviews were conducted with the help of an interpreter. In later years of the study, interpreters were cancelled if participants demonstrated increased Norwegian skills by understanding and responding to our questions and comments in Norwegian before the interpreter had time to translate.

The participants were assessed with quantitative and qualitative methods, including symptom checklists and quality of life questionnaires, structured interviews assessing personality functioning and attachment history (Rorschach and Adult Attachment Interview), and open-ended qualitative interviews about past and present aspects of their lives. As recommended by

Mollica et al. [58], the checklists and questionnaires were completed in the presence of the researchers, who offered assistance and explanations when needed, and who listened and responded in an atmosphere of empathy. Most of the participants preferred that researchers read the questionnaire and checklist items to them and noted their responses.

At inclusion into the study, interviews were conducted over 5–6 sessions (7–10 full hours) in order to complete all the planned parts of the assessment. The appointments to the research interviews were free of charge, while payment for therapy sessions in the treatment unit followed ordinary rules. In agreement with the participants, the treatment unit was informed about the results of the study intake assessment. Number of research follow-ups depended on the participants' length of treatment and their continued response to and acceptance of invitations to participate in research interviews after termination. Mental health symptoms (symptoms of PTSD, anxiety and depression) and quality of life (QOL, four domains) were thus recorded several times, about yearly or bi-yearly, and then more infrequently, before, during, at the end of, and after therapy. The participants were included into the study over a three-year period (2006–2009), and follow-up assessments were performed at varying times and frequencies between the participants. At each follow-up, one, two, or more meetings were required, depending on the respondent's health status, extensiveness of their responses, and the use of an interpreter. Therapists were also interviewed about their professional background, their experience working with refugees, and about the specific therapy process with the patient participating in the study.

## Treatment

Treatment consisted of psychological therapies of various lengths and theoretical orientations, including trauma-directed and psycho-social interventions, and was provided mostly by specialists in clinical psychology (58.8%) and psychiatry (25.5%), in the outpatient clinics also by a few psychologists and medical doctors in specialist training, and a couple of social workers and nurses with specialization in psychotherapy. The therapies in the outpatient clinics mostly had a one to two year limit, while therapists in independent practices could offer longer-term psychotherapy. The longer lasting psychotherapies in this study were mainly relational in orientation, working with alliance, relational safety, and with the traumatic experiences themselves. Each patient had to pay co-payments adding up to equivalent to 240 Euro at the beginning of each year of treatment. The rest of the year treatment was free of charge.

## Instruments/measures

The Harvard Trauma Questionnaire -Trauma Events (HTQ, Part 1; [58]), is a checklist of potentially traumatic experiences (PTEs) related to war, persecution, and other human rights violations (HRVs). The revised Cambodian version was modified, as recommended by Mollica et al., to encompass the PTEs most relevant to the group in question–here, a multinational sample. Our list thus consisted of 37 different types of PTEs related to war and HRVs such as armed attacks, persecution, and lethal danger, killings of loved ones, imprisonment, and torture. There were no age requirements to scoring of these experiences. Each confirmed type of experience is scored 1.

The Harvard Trauma Questionnaire, Symptoms of PTSD (HTQ, Part IV, Questions 1–16) [58], is a symptom checklist consisting of 16 items derived from the criteria for PTSD in the Diagnostic and statistical manual of mental disorders, fourth edition (DSM-IV) [60], which was the edition in use when this study started. The presence and severity of each item/symptom is scored from 1 (*not at all*) to 4 (*extremely*). Mollica et al's [58] validation study resulted in a clinical cutoff of $\geq 2.5$, with a sensitivity of 78% and a specificity of 65% for PTSD.

Scores ≥ 2.5 were considered checklist positive for PTSD. The mean of all 16 items (PTSD-Total), and the means of the three symptom clusters; PTSD-Reexperiencing (Questions 1, 2, 3, and 16), PTSD-Arousal (Questions 6, 7, 8, 9, and 10), and PTSD-Avoidance (Questions 4, 5, 11, 12, 13, 14, and 15) were used in the analyses.

The Hopkins Symptom Checklist-25 (HSCL-25 [58]) consists of 10 items tapping symptoms of anxiety (non-specific) and 15 items tapping symptoms of depression. The presence and severity of each item/symptom is scored from 1 (*not at all*) to 4 (*extremely*). Mean scores above 1.75 on the 10 items tapping anxiety and on the 15 items tapping depression was found consistent with clinically significant anxiety (generalized anxiety disorder) and major depression, respectively, with high sensitivity and specificity. Clinical cutoff was thus set to > 1.75. [58]. According to Mollica et al. [58] the HTQ and the HSCL- 25 were not designed as a self-report, but should always be administered to traumatized refugees by health care workers, in an atmosphere of empathy.

The World Health Organisation Quality of Life Bref (WHOQOL-Bref) [59, 61] consists of 26 items organized into Domain 1, Physical Health (7 items); Domain 2, Psychological Health (6 items); Domain 3, Social Relationships (3 items); and Domain 4, Environment/Environmental Conditions (8 items). Each item is scored on a Likert scale from 1 (*not at all/very dissatisfied/ disagree strongly*) to 5 (*completely/very satisfied/agree strongly*). Following the most commonly used option in the WHOQOL-Bref Manual, raw scores were converted to a scale from 0 (*very poor*) to 100 (*excellent*). A worldwide field study among rich and poor, sick and healthy individuals by Skevington et al. [61] resulted in the following means: 64.8 for Physical Health, 60.0 for Psychological Health, 57.2 for Social Relationships, and 54.0 for Environmental Conditions. (For comparison with the initial QOL mean scores of our participants, see Table 1)

Exile life functioning was represented by the variables *Communicated in Norwegian*, tapping whether or not Norwegian language competence was sufficient to take part in the research interviews in Norwegian, and *Employed or Studying*, tapping whether or not the participants held employment or were engaged in formal studies, even if on sick-leave at the time of assessment. Both were dichotomously scored 1 (yes) or 0 (no), and expressed by the percentage of participants scoring yes. The scoring of the variable *Communicated in Norwegian* was based on our experience in the interviews. This was a crude measure. Without our presence, explanations and examples, fewer would have been able to accurately understand and respond to the questionnaire items. The participants' employment or student status was based on their report, and was merged into one variable. We see exile language acquisition, ability to work and study as partly dependent on a tolerable level of anxiety and intrusive thoughts, and thus possible signs of increased mental well-being, as well as prerequisites to function as integrated members of the society.

The number of therapy sessions and treatment length in years, were calculated from information in the treatment records and from therapists' reports in our interviews with them. When there was more than one treatment episode during the study, the sessions and months in each of the treatments were added.

## Statistical procedures

Descriptive analyses were included for research participation, questionnaire completion, and treatment dose. The outcomes studied during follow-up included posttraumatic stress symptoms, anxiety and depression symptoms, and QOL in four domains assessed through the 10 years of follow-up. Also included were two areas of exile life functioning, the variables Communicated in Norwegian (ability to take part in the research interviews in Norwegian) and

Employed or Studying (holding employment or a student status). Changes from intake to the situation at the last research contact were investigated by exact McNemar tests. To examine changes in mental health symptoms and QOL, including all measures obtained at different times and intervals for the 54 participants over the 10 years of study, we used linear mixed effects (lme). lme is a statistical method suited to analyse repeated measurements in longitudinal studies, even when number of observations and the observation times differ among subjects [62, 63]. Time was modelled as continuous from T1. All measures over the entire research period were utilized to estimate the progress. Missing data were left missing. Random effects were included for intercept and slope, and the random structure was simplified as recommended by Pinheiro and Bates [63], if confidence intervals for random effects were not estimable or confidence limits were severely small or large. Thus, for QOL Environmental Conditions, the random structure had to be simplified by including random variation for the intercept only. Analyses of changes in symptoms of PTSD, anxiety, depression and QOL over time were performed. Marginal $R^2$ was computed as a measure of effect size for general linear mixed effects models by a method developed by Nakagawa and Schielzeth [64]. Marginal $R^2$ is defined as the proportion of variance attributed to the fixed factors, compared with the sum of variances for random effects. Further, we computed an effect size measure $r$ defined as the square root of the marginal $R^2$. Cohen's [65] benchmarks were used to evaluate the effect sizes: a correlation coefficient $r = .10$ represented a small effect, $r = .30$ a medium effect, and $r = .50$ a large effect. Descriptive statistics and t-tests were analysed by IBM SPSS Statistics, version 26. Mixed effects models were analyzed by the R package nlme and exact McNemar tests by the R package exact 2x2 [66].

## Results

### Descriptives

Fifty-four participants met for the intake interviews. One of them did not wish to respond to the questionnaires at intake, but did so at later follow-ups. During the first three years of follow-up, we only lost three participants from the study; and among these three, we were able to obtain information on language and work status for one of them about three years after intake [57]. Almost three fourths (74.1%) of the participants chose to continue taking part in the study after three years, when participants were asked for a renewed written consent. After this time, research participation dropped little by little, usually some time after therapy termination. When data collection was terminated in 2017, participants' total time taking part in the study was from 1 year up to 10.9 years (We have rounded down to 10 years in description of the study because only two participants took part in the study more than 10.49 years. Mean duration of participation in the study was 5.2 years (SD = 2.7). One third of the participants (33.3%) took part in the study for six or more years. One participant met with us only at one assessment-point, seven participants met only at two assessment points, 41 participants met with us at three to five assessment-points, and 5 participants met with us at six to seven assessment points. Over the entire study, the researchers spent from about 12 to 24 full hours interviewing each participant.

Questionnaire completion: Over the years, 54 participants completed altogether 201 HSCL-25 questionnaires, tapping anxiety and depression, varying from 1 to 7 times each between the participants, with a mean of 3.7 times each. Due to lack of endurance, time restraints and prioritizing of the qualitative aspects of the main study, the participants responded slightly less to the other questionnaires; 54 participants completed 160 HTQ Part IV questionnaires (varying from 1 to 7 times, with a mean of 3.0 times each), tapping symptoms of PTSD, and 53 participants completed 156 WHOQOL-Bref questionnaires (varying from 1 to 7 times, with a mean

of 2.9 times each), tapping four domains of quality of life. The follow-up assessments over the 10 years of study were obtained at different times and frequencies, i.e. were not synchronized between participants, and can therefore not be shown in a table.

Among the refugee patients, 34 started in the outpatient mental health clinics, and 20 in individual psychotherapy practices with public funding. For some participants, research follow-up some time after termination of treatment in the outpatient clinics showed that participants were still greatly suffering, and often they were not admitted for further therapy in the outpatient clinics. The researchers were then able to allocate some to individual practitioners. This included six of the participants.

Mean length of the participants' therapies during the whole study period was 61.3 sessions (*SD* = 74.5) over 3.4 years (*SD* = 3.1). Among the participants, 31.5% had 1 year of treatment or less, 22.6% had 6 years of treatment or more. Furthermore, 24.1% (*n* = 13) of the participants had less than 11 sessions, 38.9% (*n* = 21) had more than 45 sessions, and 25.9% of the participants had 100 or more sessions. The few patients terminating therapy after only a few sessions had various reasons for this. In a qualitative interview, one of these patients told that he terminated his contact with the therapist in the specialist service, as he felt sufficiently helped by the T1 research interviews. A couple of patients terminated prematurely for economic reasons, but took part in research interviews, which were free of charge. Some of the therapists, especially in the "private" practices, were able to keep on working with their patients for several years, in the hope that a fuller recovery would be possible. An ad hoc computation of the relationship between length of research contact and length of treatment showed that participants with longer treatments also took part in research follow-ups over a longer period of time (*r* = .536, *p* < .001).

## Changes in mental health and quality of life

The mixed effects analyses showed that over time, the participants improved significantly in symptoms of PTSD, anxiety and depression, and in three aspects of QOL. Among them, significance ranged from *p* ≤ .001 to *p* = .030, and effect sizes ranged from *r* = .05 to *r* = .38. The effect sizes were small for all the symptom changes (*r* = .05 to .13) and for the QOL variables Social Relationships (*r* = .07) and Environmental Conditions (*r* = .13), while the changes in the QOL variables Psychological Health (*r* = .38) and Physical Health (*r* = .32) were of medium effect size (see Table 2).

The lme modelled linear trajectories. Non-linearity was checked, but there were no clear indications of non-linear trajectories (see S1 Fig). At intake, 78.9% (*n* = 41) qualified for a PTSD diagnosis, 96.2% (*n* = 51) for an anxiety diagnosis, and 98.1% (*n* = 52) for major depression, while at the last meeting with each of the participants, 59.3% (*n* = 32) qualified for a PTSD diagnosis, 81.5% (*n* = 44) for an anxiety diagnosis, and 85.2% (*n* = 46) for major depression, according to the clinical cutoff of the symptom checklists (see Table 1, S1 and S2 Data).

The trajectories of symptom improvement in PTSD Total, PTSD Reexperiencing, PTSD Arousal, PTSD Avoidance, Anxiety, and Depression are illustrated in Fig 1.

The participants' improved quality of life is illustrated by the trajectories of the four domains in Fig 2.

## Changes in exile life functioning

The last time each of the 54 participants met with us, 9 more participants (5 females, 4 males) could communicate in Norwegian during interviews, and 8 more of the participants (2 females, 6 males) had gotten employed or started studying during follow-up. However, one participant stopped working due to increasing mental health problems. Thus, the number of

**Table 2. Changes in symptoms and quality of life over the 10-year follow-up.**

| | Estimate (95% CI) | p | PP | t (df) | R² | r |
|---|---|---|---|---|---|---|
| *Symptoms*: | | | | | | |
| **PTSD-total** | -0.03 [-0.04, -0.01] | .003 | -0.87 | -3.03 (105) | .003 | .05 |
| **PTSD-Reexperiencing** | -0.03 [-0.05, -0.01] | .004 | -0.93 | -2.94 (105) | .003 | .05 |
| **PTSD-Arousal** | -0.03 [-0.05, -0.01] | .001 | -1.08 | -3.28 (105) | .006 | .08 |
| **PTSD-Avoidance** | -0.02 [-0.04, -0.00] | .030 | -0.70 | -2.21 (105) | .007 | .08 |
| **Anxiety** | -0.03 [-0.04, -0.01] | < .001 | -0.97 | -3.77 (146) | .016 | .13 |
| **Depression** | -0.03 [-0.05, -0.01] | < .001 | -0.97 | -3.50 (146) | .006 | .07 |
| *Quality of Life*: | | | | | | |
| **Physical Health** | 0.96 [0.50, 1.41] | < .001 | 0.96 | 4.18 (102) | .102 | .32 |
| **Psychological Health** | 1.19 [0.67, 1.71] | < .001 | 1.19 | 4.53 (102) | .148 | .38 |
| **Social Relationships** | 0.30 [-0.42, 1.01] | .411 | 0.30 | 0.83 (102) | .005 | .07 |
| **Environment** | 0.42 [0.05, 0.79] | .025 | 0.42 | 2.28 (102) | .017 | .13 |

Linear mixed effects analyses including results from 160 completed HTQ-PTSD symptom checklists, 201 completed HSCL-25 anxiety and depression checklists and 156 completed WHO Quality of Life questionnaires from the 54 participants over the research period. Estimate (fixed effects coefficient) = difference in outcome per time unit (= 0.5 years). PP = Estimate transformed to percentage points (100 x estimate/range of the outcome scale). R2 was computed by Nakagawa and Schielzeth's method [64] for obtaining marginal R2 from generalized linear mixed effects models. r (effect size) = square root of the marginal R2. Negative values for mental health symptoms and positive values for quality of life represent improvement.

participants who could communicate in Norwegian increased significantly (p = .004) from 50.0% (n = 17) to 66.7% (n = 36), and the number of participants who were engaged in paid work or formal studies increased significantly (p = .039) from 22.2% (n = 12) at intake to the study, when all of 12 who held work were on sick-leave, to 35.2% (n = 19) at our last meeting.

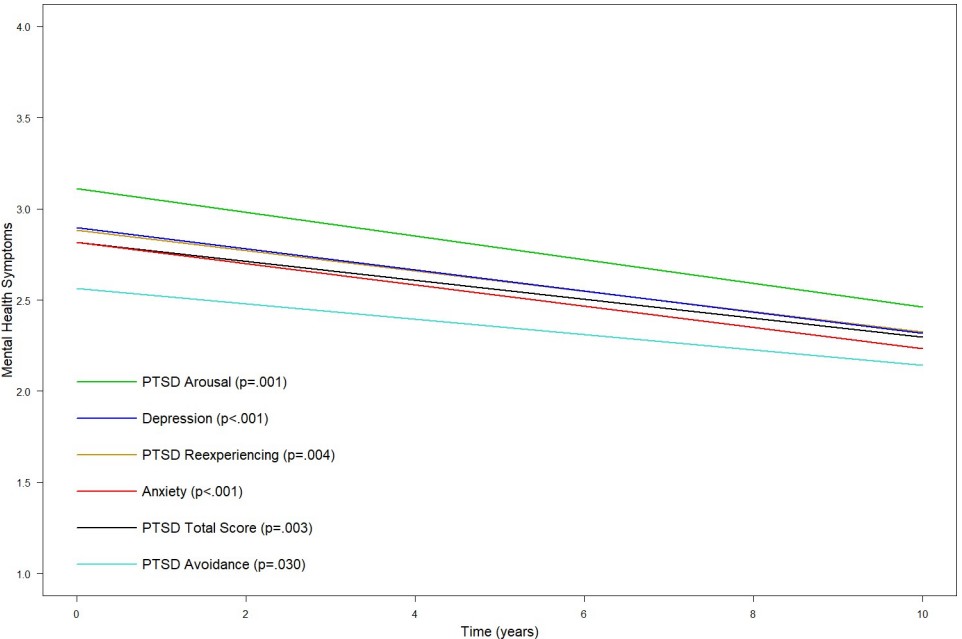

**Fig 1. Improvement in mental health symptoms over time.** All reductions in symptoms were significant. Symptoms of PTSD were measured by the Harvard Trauma Questionnaire (HTQ Part IV) [63]. Anxiety and Depression were measured by the Hopkins Symptom Checklist (HSCL-25) [63]. The plot illustrates the trajectories over 10 years from study inclusion/treatment start.

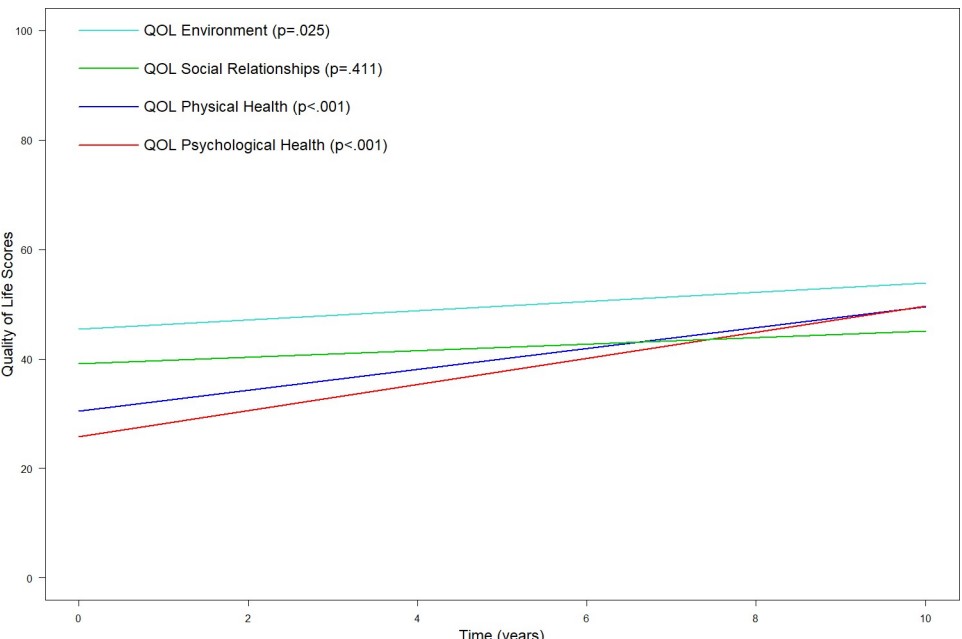

**Fig 2. Improvement in quality of life over time.** Improvements in Quality of Life were significant in the domains Physical Health and Psychological Health ($p < .001$), and Environment ($p = .025$). Quality of Life: WHOQOL-Bref [65]. The plot illustrates the trajectories over 10 years from study inclusion/treatment start.

None of them were at this point on sick-leave. A supplementary independent samples t-test of the difference in "Length of stay in Norway at our last meeting" between those who changed from not being able to communicate in Norwegian at the beginning of the study to being able to communicate in Norwegian at our last meeting, and those who did not change status concerning language competence, revealed a significant difference of about six years ($p = .003$), indicating that the shorter the stay in Norway, the greater the likelihood of a change from not being able to communicate in Norwegian to being able to do so. A corresponding supplementary independent samples t-test of the difference in "Length of stay in Norway at our last meeting" between those who changed from not being employed or enganged in formal studies at the beginning of our study to having gotten employed or entered formal studies at our last meeting. revealed no significant difference ($p = .598$). However, an ad hoc exact chi-square test showed that 44.4% of those who learned to communicate during the study, also got employed or started in formal studies, and 50% of those who got employed or started in formal studies had also learned to communicate in Norwegian during the course of the study ($p = .033$).

## Discussion

With repeated interviews and assessments over 10 years, this study examined the mental health trajectories of 54 (former) patients with refugee background and trauma related mental health disorders. The research interviews started at the time of the participants' entrance into psychological treatment, continued through their time attending treatment, and followed up through variable lengths of time after termination. The therapies were of differing lengths and theoretical approaches, but all addressed to some extent the patients' traumatic experiences. The mean duration of the therapies was 61 sessions over about three years. Although most therapies were planned as weekly, the average, holidays deducted, turned out to be roughly two sessions per month. According to the therapists, some patients cancelled sessions for physical health

reasons, some to follow family members to various appointments, and some were absent without notice due to having forgotten sessions or overslept after sleepless nights.

The linear mixed effects analyses including measurements over the entire research period, demonstrated that the participants' long-time improvement, at the mean level, was significant but modest in symptoms of PTSD, anxiety and depression. Moreover, the participants improved significantly, and more decisively, in subjectively felt quality of life in the psychological and physical health domains. The linear mixed effects results may not be reliable during late follow-up since only a small group of participants had long follow-up times. Many of these received long-term therapy. The substantial uncertainty during late follow-up can also be seen from the size of the grey, trumpet-like shadows on the smooth curves towards the 10-year mark [S1 Fig]. We found significant improvement in exile life functioning among the participants, in terms of the number of participants who, at the end of follow-up, could communicate in Norwegian and in the number who were employed or had entered formal studies.

The modest symptom improvement can be attributed to the protracted, severe, and complex condition of the refugee patients at treatment start. In the introduction, we reported on the chronic trajectories of many refugee patients. The mean duration of residence in Norway when entering treatment and this study, was 10.5 years. Many clinical studies, in addition to ours, have found that refugee participants have stayed for several years–with means from five up to 16 years–in the host country before entering mental health treatment [27, 67]. The late entrance to mental health services may be part of the reason for the chronicity of many of these patients' mental health disorders, despite therapy. According to the findings by Haagen et al. [49], another factor contributing to the modest improvement, may be the high level of comorbid depression at treatment start. Comorbidity and psychosocial problems in general clinical populations predict poorer treatment response, and among patients with complex presentations of PTSD, with comorbidity, and with chronic conditions at the outset of therapy, many continue to have substantial symptoms after termination [46, p. 174]. Thus, the longitudinal, small, average symptom reduction found in this study may be as good as it gets, considering the complexity and chronicity of many of these patients' mental health disorders in combination with psychosocial problems and an often marginalized position in society.

Concerning the participants' mental health changes, symptoms of anxiety decreased the most, then the arousal and avoidance clusters of PTSD, thereafter depression. The reduction in symptoms was lowest within the reexperiencing cluster of PTSD. This implies that even if the participants were still bothered by recurring thoughts related to the traumatic events, reexperiencing did not evoke as much anxiety. The decrease in arousal and avoidance can be interpreted in the same direction: trauma memories persist, or are easily evoked, but the participants do not engage in as much avoidance behaviour, and there is not as much evidence of bodily or emotional upheaval. The decrease in depression can be understood as a sum effect of general symptom improvement and improved functioning. Even if the changes are of small effect, these seem to be important results. However, as noted by Lambert [46], symptom reduction may not be the most important sign of improvement with severely traumatized patients and may not be the best way to evaluate outcome. Furthermore, for this patient group, a full recovery in terms of symptom removal may not be a realistic treatment goal. The greater effect size of improvement in quality of life relative to symptom reduction among the participants in this study could indicate that the participants coped with their symptoms with greater strength or less suffering.

The participants' perceived quality of their psychological health improved the most, but was closely followed by their subjectively felt quality of physical health. The joint initial low level of, and subsequent joint improvement in the participants' experience of the quality of their psychological and their physical health indicates the close relationship between physical

and psychological well-being, and resonates well with the finding by e.g. Berthold et al. [21], of the connection between physical and mental health problems among refugees. The participants' perceived quality of their environmental conditions improved somewhat less, but then their initial score was not as low as other aspects of quality of life at intake to the study–"probably owing to the relatively good welfare system in Norway" [57, p. 40]. Finally, the participants' subjective experience of their quality of social relationships also improved, although less than other aspects of quality of life. Our impression from the qualitative interviews was that many of our informants struggled with relating socially to anyone outside the immediate family. A separate study of children in refugee families, including several children of participants in this study, showed that the children took much responsibility for parents' social function, including relations outside the family [68, 69]. Among the adult participants in the present study, however, some appeared to become more socially confident after the benign experience of talking with their therapist, and when the pressure of their trauma-related and depressive ruminations decreased.

The significant increase, during the study, in the number of participants who could communicate in Norwegian during interviews, was, according to the supplementary analysis, partly related to being newer in the country, so that some more years of stay made a positive difference. Other not-investigated factors must also have been of importance to the result. The increase in the number of participants who were employed or engaged in formal studies was also significant. Here, the supplementary analysis showed no significant association between change in job/study status and number of years spent in Norway. It may be worth noting that in addition to the relatively small increase in participants who became employed or started studying, all except one of those who already held employment (although on sick-leave) when they entered treatment and the study, managed to get back to work and stay employed. Our belief is that therapy was important to take the step into employment or studies, as well as recuperating and holding onto jobs. Considering the participants' mean age of nearly 40 years, their mean decade long stay in Norway, and their likely chronic mental health condition at entry to therapy and the study, the results are relatively encouraging. The improvement in exile life functioning together with the improved quality of life, may demonstrate that even small effect changes in symptoms can have important consequences for the participants' lives. According to Schick et al.'s [31] proposal of a circular association between mental health, well-being, and exile life functioning, positive change in any of these areas may also fuel improvement in other areas.

Not as easily measured as items on a checklist, but supported by our impressions from the qualitative interviews, we believe that the contact over time with a benign, empathising and interested therapist gives hope, alleviates stress, encourages the patient to move on in life, and thus contributes to increased tolerance of symptoms, greater well-being, and improved functioning.

In the participants' therapies, the number of therapy sessions was not set in advance, but was dependent in part by each participant's needs and course of improvement. In the mental health outpatient clinics participating in this study, few patients were allowed more than 1–2 years of treatment. The individual practice specialists, however, could work with their patients over several years. The results showed that about one fourth of the participants had less than 11 therapy sessions, a non-therapeutic number of sessions according to the studies of Lambert and of Hansen et al. [47, 48]. On the other hand, almost 39 percent of the patients had more than 45 sessions, the minimum number of sessions that were estimated to be required for improvement among patients with comorbid disorders and psychosocial problems [47, 48]. In our study, nearly 26 percent of the participants had 100 or more therapy sessions, a number of sessions that will only infrequently be offered to refugee patients in Norway, and probably also

in most other countries. We have not examined the relationship between therapy dose (number of sessions) and improvement in mental health symptoms and QOL in the present study. In general, there is little correlation between number of sessions and outcome, as discussed in the introduction [46]. However, from the qualitative interviews with patients and with their therapists, we could see that for some patients, the long-term therapy over several years resulted in great steps forward, while for others, their suffering seemed to be as severe when the therapy was terminated. Still, one of the patients, who after many years of therapy seemed not to have progressed, said to his therapist in the last session: "You have helped me much more than you think". In line with De Smet et al. [50] we suggest that discerning what constitutes good treatment and beneficial outcome for this patient group warrants closer study.

The changes observed in this study may be attributed to a combined effect of the personal qualities of the participants, the therapy experience, including the therapist's personal qualities, therapeutic style, and interventions, the match between therapist and patient, the comprehensive research interviews, experienced by many of the participants as therapeutic in themselves, life events outside therapy, and the passing of time. We tentatively refer to the previously offered suggestion [57, p. 56], that the long stay in Norway, the longevity of functional impairment, and the severe load of anxiety, depression and posttraumatic stress symptoms upon entering therapy, makes it unlikely that significant improvement during the next years was due to time alone.

The significant improvements found over the entire 10 years of data collection, indicated that gains obtained at the three-year follow-up [56] were more or less maintained. However, we observed extensive individual differences in treatment response/long-term outcome among the participants at the end of the study, and many of the participants still had a high symptom burden. Therefore, predicting who in refugee patient groups can benefit from therapy in general, and long-term treatment in particular, may be difficult. Nevertheless, leaning on a conclusion by Bager et al. [70] about the positive family level cost-benefit effect of psychosocial rehabilitation for severely traumatized refugees, we consider it well worth the effort to invest in the wellbeing of the individual patient, with the anticipated spin-off effect upon the patient's spouse and children.

## Strengths and limitations

This study, with its longitudinal design including several outcome measures, may contribute to knowledge regarding the long-term course during and after psychological treatment, of traumatized refugees' mental health, well-being and functioning in exile. The broad inclusion criteria and the high acceptance rate in the study imply that the participants can be seen as representative of the clinical population of trauma-affected refugees being accepted for psychological treatment in and around the Oslo area of Norway at the time of inclusion. The patients received treatment of varying theoretical orientations and lengths, according to the patients' needs, the therapists' appraisal of appropriateness and effect, and the policy of the treatment units.

A limitation of the study is that we did not perform a pre-study power calculation in 2006. Furthermore, without a controlled design, we cannot claim that any positive changes can be attributed to the treatment received. However, the participants' mean stay in Norway was more than 10 years when they entered the study, indicating increased or continued suffering rather than improvement over time. Likewise, the long duration of follow-up, with repeated measurements, and the statistical method employed, which made use of all the measuring points, speak against the notion that we might have captured a top in suffering at inclusion, and at later measurements a bottom of the wavelike ups and downs of mental health suffering

found in refugee studies. A limitation is that using linear mixed effects analysis did not allow us to see when changes appeared, as they were modelled linearly. The great variation in national origins, time of stay in Norway, length and type of therapy, and number and frequencies of research follow-ups, limit what analyses can be made and what conclusions can be drawn. We have not yet examined the trajectories of different subgroups among the participants, for example of the third among the participants who continued to take part in research assessments the last few years. Looking into this subgroup could inform us whether they were among cases with less or more improvement, causing the long-term mental health trajectory to inflate or deflate.

The participants did not respond to all the questionnaires at each point in time. We may have wanted to do too much at each research meeting with the refugees. Due to the participants' many strains of living, their difficult social situation, and their mental and physical problems, many of the participants did not have the time or strength to respond to all of the questionnaires and the semi-structured qualitative interview questions each time. Many preferred to talk freely about their life situation and experiences.

## Conclusion and implications

During this 10-year follow-up study, the traumatized refugees had attended psychotherapy of varying length and content in publicly funded, specialist mental health services in Norway. The participants improved significantly and modestly in mental health symptoms, but more in experienced quality of psychological and physical health. Thus, how one lives with ones symptoms may be more important than the level of symptoms themselves. Some were able to start formal education, some who had not previously been employed, started working, and some learned to communicate in Norwegian during the course of the study. The results hide substantial individual variation in improvement. As evidenced in the quantitative data and our repeated meetings with the participants, some benefited to a great extent, others moderately, and a few, left untouched by the therapy experience, just continued on a downwards spiral. Discerning which characteristics impact on the degree of benefit from psychotherapy, and which barriers may be at play for those who do not seem to benefit, is vital to be able to help as many of these patients as possible and to use existing resources in a good way. Major symptom reduction may not be attainable, and may not be the most important indication of long-term improvement among refugees with long-standing and trauma-related suffering. In addition to symptoms mitigation, evaluation of treatment effects should take into consideration a broad array of outcomes, including physical and psychological well-being, exile life functioning, relational security, and social network. Furthermore, individuals or subgroups should be studied in detail to learn more about factors related to change.

Preventive efforts and mental health interventions at an earlier time, as in a stepped care approach [71, 72], could counteract the chronicity of mental health suffering in this at-risk group. For refugees in need of mental health treatment, we suggest that working simultaneously with mental health problems, host language acquisition, and employment may increase gains and benefit all of these areas. Finally, in accordance with Hynie [29] we propose that the whole person, embedded in his or her familial and societal context, and examination of ongoing stress factors that need to be alleviated, should be included in approaches to promote health and restore functioning for traumatized refugees.

## Supporting information

**S1 Fig. Smooth curves to check for non-linearity.**
(PDF)

**S1 Data. HTQ PTSD datafile.**
(SAV)

**S2 Data. HSCL-25 anx_depr datafile.**
(SAV)

**S3 Data. WHOQOL qol datafile.**
(SAV)

**S4 Data. LanguageWorkStudy datafile.**
(SAV)

## Author Contributions

**Conceptualization:** Marianne Opaas, Sverre Varvin.

**Data curation:** Marianne Opaas.

**Formal analysis:** Marianne Opaas, Tore Wentzel-Larsen.

**Investigation:** Marianne Opaas, Sverre Varvin.

**Methodology:** Marianne Opaas, Tore Wentzel-Larsen, Sverre Varvin.

**Project administration:** Marianne Opaas, Sverre Varvin.

**Visualization:** Marianne Opaas, Tore Wentzel-Larsen.

**Writing – original draft:** Marianne Opaas.

**Writing – review & editing:** Marianne Opaas, Tore Wentzel-Larsen, Sverre Varvin.

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
