## [Decision Letter · Decision Letter 0]

23 Jul 2020

PONE-D-20-12948

The 10-Year Course of Mental Health, Quality of Life, and Daily Life Functioning in Traumatized Refugees from Treatment Start

PLOS ONE

Dear Dr. Opaas,

Thank you for submitting your manuscript to PLOS ONE. After careful consideration, we feel that it has merit but does not fully meet PLOS ONE’s publication criteria as it currently stands. Therefore, we invite you to submit a revised version of the manuscript that addresses the points raised during the review process.

The reviewer has identified several important points that you must address in order to meet the publishing criteria of PLOS ONE, and to improve the clarity and replicability of your research.

We look forward to receiving your revised manuscript.

Kind regards,

Melita J. Giummarra

Academic Editor

PLOS ONE

Journal Requirements:

2. Please provide additional details regarding participant consent. In the ethics statement in the Methods and online submission information, please ensure that you have specified (a) whether consent was informed and (b) what type you obtained (for instance, written or verbal, and if verbal, how it was documented and witnessed). If the need for consent was waived by the ethics committee, please include this information.

Reviewers' comments:

Reviewer's Responses to Questions

**Comments to the Author**

1. Is the manuscript technically sound, and do the data support the conclusions?

Reviewer #1: Partly

2. Has the statistical analysis been performed appropriately and rigorously? 

Reviewer #1: No

3. Have the authors made all data underlying the findings in their manuscript fully available?

Reviewer #1: No

4. Is the manuscript presented in an intelligible fashion and written in standard English?

Reviewer #1: Yes

5. Review Comments to the Author

Reviewer #1: Thank you for an interesting study regarding the course of mental health, quality of life, and daily functioning inf traumatized refugees. This manuscript is very long and repetitive (e.g. multiple times you discuss the study details) such that the reader tends to get lost in the detail. It would be beneficial to refine the introduction to be more concise and focussed on the main aim of this study and reduce the repetition in the Methods. In addition, there are Results included in Methods such that it is unclear to the reader when your results section commenced.

Below are some specific comments:

1. Introduction: I found the introduction, whilst extremely interesting, it was too long and needs to have a clearer path towards the overall aim of this paper. For example, lines 86 to 151 seems to be discussing therapies, but your participants seemed to be provided with a wide variety of treatments from a variety of specialists (e.g. lines 277 to 286). It is unclear that your models measured the nature of the treatment?

2. Sample size: Please perform a power analysis for your mixed methods models (Note: there are a number of R packages that can perform these for you)

3. Participants: Please report on the proportion from Asia, Europe and Africa as these continents do differ and the reader should understand the mix of your participants

4. Procedures: It is unclear as to the distribution of timing of termination of therapy as in line 265 you state “…. Termination of therapy, whenever that occurred”.

5. Measures: Please provide citations for your clinical cutoff. Also, please ensure you outline to the reader what high scores of your measures mean. Please clarify if these are self-report measures as you reference the DSV-IV but it does not appear to be SCIDs?

6. Measures: Please provide the relevant research behind your development of the “Daily life function”

7. Missing Data: Please report on the missing data on both the time invariant and time varying variables.

8. Table 2: Please include base at each time point and for each measure. Also, your title indicates this is a 10-year study, but Table 2 only reports up to the first 3 years. Please provide the relevant information.

9. Diagnostic Evaluation: (Lines 339-350) It is very unclear who coded the ICD-10 diagnoses and at what time period(s) these diagnoses occurred? This is the first time ICD-10 has been mentioned and requires more clarification as these do not appear to be clinical diagnoses so it is unclear as to why they are referred to? ICD-10 is predominantly used for billing purposes and within hospitals for general diagnoses and procedure reporting. Also, please define “ICD-10” to the reader.

10. Statistical Procedures: You indicate you modelled time “as continuous from T1” (Line 363) but this assumes equal time periods but you have T1 baseline, 1 year, then 3 years, then it is unclear the other time periods? LME assumes a linear relationship for equal increments and this does not appear to be the case for your data. Did you perform a sensitivity analysis with time as a categorical variable with T1 as base reference? If not, please do so.

11. Statistical Procedures: Please include the ICC.

12. Statistical Procedures: In your LME models, did you control for potential time invariant confounders such as continent? What interactions with time did you test?

13. Results: Please provide sample sizes for all your results as it appears much of your results are based on small sample sizes. For example, Changes in Daily Life Functioning have n<10.

14. Results: Your graphs extend to 6 years, but your descriptives are up to T3 and the title of your manuscript indicates up to 10 years. Please ensure your time points are consistent for the reader.

6. PLOS authors have the option to publish the peer review history of their article (what does this mean?). If published, this will include your full peer review and any attached files.

Reviewer #1: No

---

## [Author Response · Author response to Decision Letter 0]

10 Sep 2020

PONE-D-20-12948 PLOS ONE

The 10-Year Course of Mental Health, Quality of Life, and Exile Life Functioning in Traumatized Refugees from Treatment Start

Response to reviewers

(Our replies/comments in italics)

1. Response to academic editor 

We have rechecked PLOS ONE’s style requirements and made the ensuing corrections. I hope the style is now correct. 

2. Please provide additional details regarding participant consent. In the ethics statement in the Methods and online submission information, please ensure that you have specified (a) whether consent was informed and (b) what type you obtained (for instance, written or verbal, and if verbal, how it was documented and witnessed). If the need for consent was waived by the ethics committee, please include this information.

Under Design in the Methods section, we added the last sentences (here in italics) to the following passage: “The study was approved by the Norwegian South-East Regional Committee for Medical and Health Research Ethics [61] and therapy was conducted according to the Declaration of Helsinki: Ethical Principles for Medical Research Involving Human Subjects [62]. Participants’ written, informed consent was required to take part in the study. The first consent was limited to three years. After three years, the participants were asked if they wanted to continue taking part in the study, and those who accepted, gave a new, non-time-limited informed and written consent.” Furthermore, under Procedures in the Methods section, the following is specified (with new parts in italic): “Participants were carefully informed, in writing and verbally, about the purpose of the study, about confidentiality, that participation in the study was voluntary, that rejecting was of no consequence for their treatment, and that, if they signed up for the study, they could withdraw at any time. Seventy-two patients were asked to participate in the study, and 54 accepted and gave their written consent (75% inclusion).”

We have uploaded the minimal anonymized data set necessary to replicate our study findings as Supporting Information files. 

2. Response to reviewer

Thank you very much for your time reviewing our manuscript and for helpful comments. 

1. Is the manuscript technically sound, and do the data support the conclusions?

Reviewer #1: Partly

Answers are given below to the specific comments from the reviewer.

2. Has the statistical analysis been performed appropriately and rigorously? 

Reviewer #1: No

Answers are given below to the specific comments from the reviewer.

3. Have the authors made all data underlying the findings in their manuscript fully available?

The PLOS Data policy <http://www.plosone.org/static/policies.action#sharing> requires authors to make all data underlying the findings described in their manuscript fully available without restriction, with rare exception (please refer to the Data Availability Statement in the manuscript PDF file). The data should be provided as part of the manuscript or its supporting information, or deposited to a public repository. For example, in addition to summary statistics, the data points behind means, medians and variance measures should be available. If there are restrictions on publicly sharing data—e.g. participant privacy or use of data from a third party—those must be specified.

Reviewer #1: No

We have now uploaded the data underlying our findings as Supporting Information files.

4. Is the manuscript presented in an intelligible fashion and written in standard English?

Reviewer #1: Yes

Thanks for this positive assessment.

Reviewer’s comments (normal font) and our responses (in italics) 

1. Introduction. : I found the introduction, whilst extremely interesting, it was too long and needs to have a clearer path towards the overall aim of this paper. For example, lines 86 to 151 seems to be discussing therapies, but your participants seemed to be provided with a wide variety of treatments from a variety of specialists (e.g. lines 277 to 286). It is unclear that your models measured the nature of the treatment?

We have tried to shorten and focus the introduction in a better way, as recommended. It is true that our models did not measure the nature of the treatment. We have therefore omitted the discussion of therapy methods in the introduction. We have also omitted the part on whether treatment results in this patient group represents “value for money”, and made some additional adjustments to the introduction, hopefully representing a clearer path towards the aim of the paper. However, we have kept passages that more generally refer to results of therapy for this patient group. We have also kept passages leading up to the importance of longitudinal studies, and passages commenting on how or whether length of treatment is related to treatment outcome. Part of our participants received very long treatments, but they may not have progressed correspondingly. 

2. Sample size: Please perform a power analysis for your mixed methods models (Note: there are a number of R packages that can perform these for you). 

In 2006 we consulted two statisticians about sample size in our study. They said a sample size of 100 would be better, but they gave the opinion that with reasonable effect sizes of expected changes, and repeated measurements, we could still do meaningful analyses with an N of about 50. At that time, we did not have resources to continue including more participants to the study. Aims of the study were multiple, and part of the aims were explorative, since in-depth, longitudinal and multi-method studies of this patient group was largely non-existing at the time – and is still scarce. 

A formal power analyses was thus not performed at the beginning of our study. A power analysis after data collection is not recommended. We refer to the following from Lydersen (2019): “After the study, it is generally recommended to report an estimate and a 95 % confidence interval for the effect, as well as a p-value. Sometimes, a journal or a reviewer requests a calculation of the ‘observed power’ in addition (2). This means a statistical power calculation based on the observed effect, as if it were a future study, and reporting it as if it gives additional information about the study already performed. This is not only fundamentally flawed, but it gives no information in addition to the reported p-value” (p. 1). Ref.: Stian Lydersen (2019). Statistical power: Before, but not after! Tidsskrift for den norske legeforening (Journal of the Norwegian Medical Association). (https://tidsskriftet.no/en/2019/01/medisin-og-tall/statistical-power-not-after). This article includes further references with the same conclusion.

3. Participants: Please report on the proportion from Asia, Europe and Africa as these continents do differ and the reader should understand the mix of your participants.

The distribution of backgrounds has now been added: “The participants, 35 men and 19 women, came from 15 different countries in Asia, Europe, and Africa. Of these, 57.4% (n = 31) came from countries in the Middle East, 16.7% (n = 9) came from other Asian countries, 16.7% (n = 9) came from countries in the Balkans, and 9.3% (n = 5) came from various African countries.

4. Procedures: It is unclear as to the distribution of timing of termination of therapy as in line 265 you state “…. Termination of therapy, whenever that occurred”. 

In the process of avoiding repetitions in the manuscript, this phrase disappeared. The meaning was, however, “termination of therapy, which occurred after individually varying lengths of therapy”. 

5. Measures: Please provide citations for your clinical cutoff. Also, please ensure you outline to the reader what high scores of your measures mean. 

In the instruments section, references for the clinical cutoff is now provided for HTQ, part IV and for HSCL-25. The meaning of high scores should also now be clear in the description of the instruments.

Please clarify if these are self-report measures as you reference the DSV-IV but it does not appear to be SCIDs? 

The symptom measures are all based on the instruments used in the study, which are described in the Instruments section. The interviewers (a specialist in clinical psychology, and a specialist in psychiatry/psychoanalyst) generally read the questions/items to the participants, asked follow-up questions if the participant’s answer was unclear, and noted their responses. In the Instruments section, we have added the following: “According to Mollica et al. [58], the HSCL- 25 and the HTQ were not designed as a self-report, but should always be administered to traumatized refugees by health care workers, in an atmosphere of empathy.” 

In the Procedures section of the current manuscript, we have added the words marked in italic: “As recommended by Mollica et al. [58], the checklists and questionnaires were completed in the presence of the researchers, who offered assistance and explanations when needed, and who listened and responded in an atmosphere of empathy. Most of the participants preferred that the researchers read the questionnaire and checklist items to them and noted their responses.” 

In the first publication from the study in 2013 [55]), the procedure was outlined in more detail: “All questionnaires were administered personally by the researchers, question by question, and translated, when needed, orally, on the spot. Associations to the questions and exemplifications of the answers were allowed. We asked several follow-up questions before placing a check mark noting the intensity of the problem. These procedures were time consuming, but gave us indications of how the questions were understood and added important information elaborating the fixed choices of answers in the questionnaires. Mollica et al. (2004) recommended that the HSCL-25 and the HTQ always be administered to traumatized refugees by health care workers, in an atmosphere of listening with empathy. They stated that neither instrument was designed as a self-report.”

Reference to DSM-IV: 

The HTQ instrument is described with reference to DSM-IV in its manual. We have added the full name of the DSM-IV. In our manuscript, under Instruments, we describe the HTQ the following way: “The Harvard Trauma Questionnaire, Symptoms of PTSD (HTQ, Part IV, Questions 1-16) [58], is a symptom checklist consisting of 16 items derived from the criteria for PTSD in the Diagnostic and statistical manual of mental disorders, fourth edition (DSM-IV) [60],…”. So as not to cause any misunderstanding elsewhere in the manuscript, we removed the words “DSM-IV-based” from other references to the results of the symptom checklists.

6. Measures: Please provide the relevant research behind your development of the “Daily life function”.

In the Introduction we have added a passage about the interdependence between mastering the language of the host country, completing education or holding employment and mental health, wellbeing, and of functioning well in exile. Thanks for making us aware of this omission in the introduction. Throughout the text we have changed the expression “daily life function” to “exile life functioning”/” functioning in exile”. We found this to be more precise.

7. Missing data: Please report on the missing data on both the time invariant and time varying variables. 

Participants’ needs and (lack of) endurance had priority before our needs and plans as researchers. At some follow-ups, participants willingly talked with us on the qualitative parts of the follow-up, but refrained from, or had to leave before responding to one or more of the questionnaires. However, as questionnaires and symptom checklists were conducted in interview form, there were not many missing items within the questionnaires, once they were responded to at all. We conducted around 500 interviews during this longitudinal study, both the number of interviews and their timings varied considerably between participants. It would be too much detail if we were to outline each time we had a follow-up with any of the participants when one or more of the questionnaires were not completed. 

We have described in general terms in the Procedures section that participants were recruited consecutively over three years, and that they were interviewed at different dates and intervals. In the beginning of the Results section, under Descriptives, we have described how many times each questionnaire was in fact completed, and over what period of time.

The way the data was arranged for this analysis, allowed us to see how many times each participant completed each questionnaire, the date for each completion, and across how many years each respondent responded to the questionnaires. 

The following is now described in the Statistical procedures section,: “To examine changes in mental health symptoms and QOL, including all measures obtained at different times and intervals for the 54 participants over the 10 years of study, we used linear mixed effects (lme). lme is a statistical method suited to analyse repeated measurements in longitudinal studies, even when number of observations and the observation times differ among subjects [67,68]. Time was modelled as continuous from T1. All measures over the entire research period were utilized to estimate the progress. Missing data were left missing. 80 % of the items comprising each variable had to be completed if the variable was to be computed. ” 

8. Table 2: Please include base at each time point and for each measure. 

We do not fully understand this comment. Data was collected at different times and intervals. Please see our comments on point 7. 

Also, your title indicates this is a 10-year study, but Table 2 only reports up to the first 3 years. Please provide the relevant information. 

To avoid confusion of what belongs to the lme, which is the main analysis of the current study, the original Table 2 is removed and merged with Table 1 (Table 1. Demographics, mental health symptoms, quality of life, and indicators of functioning in exile upon entering therapy and the study). It now only shows results from intake – baseline. We have omitted data from T2 and T3, because they are not a result of the lme, but were formerly modelled for an earlier publication, before the data collection was completed. Table 1 is shown in the Methods section. Data from intake/baseline is included in, but not a result of the present study. 

Furthermore, to ease consistency, we have now changed the modelled trajectories in Figures 1 and 2 to show the whole 10-year period, and made sure the results are described and illustrated for 10 years throughout the study.

The data for the lme were organized consecutively, including results from each time each questionnaire was completed by each participant, and at what date. In the lme, for each individual, time for each follow-up questionnaire response was calculated from the study intake date to the date of each follow-up. 

9. Diagnostic Evaluation: (Lines 339-350) It is very unclear who coded the ICD-10 diagnoses and at what time period(s) these diagnoses occurred? This is the first time ICD-10 has been mentioned and requires more clarification as these do not appear to be clinical diagnoses so it is unclear as to why they are referred to? ICD-10 is predominantly used for billing purposes and within hospitals for general diagnoses and procedure reporting. Also, please define “ICD-10” to the reader.

All Norwegian mental health clinics, part of the Norwegian specialized health care system, use ICD-10 (The International Statistical Classification of Diseases and Related Health Problems, 10th Revision, WHO, 1994) as their formal diagnostic system. Maybe reporting of the diagnostic evaluation from the clinics only serves to confuse. To make the manuscript more readily understandable and straight forward, we have omitted information about what diagnoses the participants were given in the treatment unit. 

10. Statistical procedures: You indicate you modelled time “as continuous from T1” (Line 363), but this assumes equal time periods but you have T1 baseline, 1 year, then 3 years, then it is unclear the other time periods. 

In the Results section, under Descriptives, and in the Methods section, the variability in number of follow-ups and variability of intervals between follow-ups among the participants is described. Please also see our response to point 8.

LME assumes a linear relationship for equal increments and this does not appear to be the case for your data. Did you perform a sensitivity analysis with time as a categorical variable with T1 as base reference? If not, please do so.

Mixed effects models have the advantage of not assuming equal increments, and the measurement times may vary between participants. The lme modelled linear trajectories of the scores of the symptom clusters and quality of life dimensions. All model assumptions are approximations, but we have now checked non-linearity without finding indications of systematically non-linear trajectories. As illustrations, plots including smooth curves for PTSD, anxiety, depression and the quality of life dimensions are included as Supplemental files. As we have described, and as can be seen on the smooth curves, the results are more uncertain for the last few years, as these included responses from only a few participants, potentially a selected part of the sample. 

11. Statistical procedures: Please include the ICC. 

Our mixed effects models include random variation between persons both for the intercept and the slope, therefore an intraclass correlation is not available based on these models. 

12. Statistical Procedures: In your LME models, did you control for potential time invariant confounders such as continent? What interactions with time did you test?

There are widely different cultures represented in each continent, so in our view, it will not make sense to control for continents. Moreover, the distribution of participants is not equally dispersed among the continents. This study is based on eligible participants, without regard to national origins, among patients being referred to, and accepted for treatment, as naturally occurring in the Norwegian mental health services that we collaborated with at the time of inclusion (2006-2009). We did not test any possible confounders at this point.

13. Results: Please provide sample sizes for all your results as it appears much of your results are based on small sample sizes. For example, Changes in Daily Life Functioning have n<10.

We see that the way this was formerly outlined could be unclear. In Table 1, all N’s are now within the table. Changes in Daily Life Functioning, now termed Exile life functioning, occurred only in 10 or less (n) from intake to the study until our last meeting with each participant, but the investigated N included all of the 54; the ones who changed as well as the rest of the sample who did not change status from intake to our last meeting with them, whether they at intake already held a job and could communicate in Norwegian or not. In the Results section, Changes in Exile life functioning, we have tried to state more clearly the bases for these calculations. Sample sizes throughout the study for the data included in the lme cannot be given for reasons explained under point 7, but the number of completed questionnaires are described under Descriptives in the Results section.

14. Results: Your graphs extend to 6 years, but your descriptives are up to T3 and the title of your manuscript indicates up to 10 years. Please ensure your time points are consistent for the reader.

Please see our response to point 8. We hope the manuscript will now appear consistent throughout.

---

## [Decision Letter · Decision Letter 1]

23 Nov 2020

PONE-D-20-12948R1

The 10-year course of mental health, quality of life, and exile life functioning in traumatized refugees from treatment start

PLOS ONE

Dear Dr. Opaas,

Thank you for submitting your manuscript to PLOS ONE. After careful consideration, we feel that it has merit but does not fully meet PLOS ONE’s publication criteria as it currently stands. Therefore, we invite you to submit a revised version of the manuscript that addresses the points raised during the review process.

Thank you for your thorough attention to reviewer comments on the original manuscript. Please attend to the additional comments of the statistical review.

We look forward to receiving your revised manuscript.

Kind regards,

James Curtis West, M.D.

Academic Editor

PLOS ONE

Reviewers' comments:

Reviewer's Responses to Questions

**Comments to the Author**

1. If the authors have adequately addressed your comments raised in a previous round of review and you feel that this manuscript is now acceptable for publication, you may indicate that here to bypass the “Comments to the Author” section, enter your conflict of interest statement in the “Confidential to Editor” section, and submit your "Accept" recommendation.

Reviewer #1: All comments have been addressed

Reviewer #2: (No Response)

2. Is the manuscript technically sound, and do the data support the conclusions?

Reviewer #1: Yes

Reviewer #2: Partly

3. Has the statistical analysis been performed appropriately and rigorously? 

Reviewer #1: Yes

Reviewer #2: Yes

4. Have the authors made all data underlying the findings in their manuscript fully available?

Reviewer #1: Yes

Reviewer #2: Yes

5. Is the manuscript presented in an intelligible fashion and written in standard English?

Reviewer #1: Yes

Reviewer #2: Yes

6. Review Comments to the Author

Reviewer #1: Thank you for answering my review questions, clarifying your research methodology and arguing your stance.

Reviewer #2: A 10-year longitudinal study was conducted to study the changes in symptoms of posttraumatic stress, anxiety and depression in refugee participants receiving psychotherapy. Linear mixed effect models were used to determine changes in quality of life measures. Symptoms, quality of life, and exile life functioning improved over the course of the study.

Minor revisions:

1- There are procedures/macros available outside of the mixed effects models which can be used to calculate intraclass correlation coefficients.

2- Indicate the covariance structure used in the linear mixed model and the criteria for selecting it.

3- Within the manuscript, state and justify the study’s target sample size with a pre-study statistical power calculation for the mixed model.

4- When data is missing, indicate the analyzed sample size.

7. PLOS authors have the option to publish the peer review history of their article (what does this mean?). If published, this will include your full peer review and any attached files.

Reviewer #1: No

Reviewer #2: No

---

## [Author Response · Author response to Decision Letter 1]

25 Nov 2020

Thank you again for reviewing our manuscript, our comments to the previous review, and for your new comments.

6. Review Comments to the Author

Reviewer #1: Thank you for answering my review questions, clarifying your research methodology and arguing your stance.

Reviewer #2: A 10-year longitudinal study was conducted to study the changes in symptoms of posttraumatic stress, anxiety and depression in refugee participants receiving psychotherapy. Linear mixed effect models were used to determine changes in quality of life measures. Symptoms, quality of life, and exile life functioning improved over the course of the study.

Minor revisions:

1- There are procedures/macros available outside of the mixed effects models that can be used to calculate intraclass correlation coefficients.

Response: The question of whether to report an intraclass correlation based on our mixed effects models is not a question of lack of software. All computations involving random effects in our mixed effects models may be done within the estimated models, with no need for external software. 

As commented in our previous reply, it is not clear what could be the meaning of an intraclass correlation coefficient, in a mixed effects model with between person random variation both in the intercept and the slope. An intraclass correlation is defined in terms of standard deviations for random effects within and between persons, in a mixed effects model with only one random effect within and one random effect between persons. In our models, there is only one random effects within persons, but two random effects between persons.

2- Indicate the covariance structure used in the linear mixed model and the criteria for selecting it.

Response: We have included a random effect for within person variation and random effects for variation in intercept and slope between persons. We have not included a further covariance structure for the error terms within persons, and we have not stated that such a structure is part of the model. The reason is that the time points, and the number of time points, varies considerably between subjects. This may be a limitation, but we do not see a possibility for hypothesizing a specific covariance structure for such a case. 

3- Within the manuscript, state and justify the study’s target sample size with a pre-study statistical power calculation for the mixed model.

Response: As commented in our replies in the previous round, there was not performed a formal sample size calculation when the study was initiated back in 2006. This is a limitation, but there is no way this could be rectified now since post hoc sample size calculations are methodologically flawed, as detailed in references given in our previous reply. We have added the lack of a pre-study power calculation to Limitations, page 28, line 599.

4- When data is missing, indicate the analyzed sample size.

Response: Sample sizes and all details of the pretreatment assessment and the assessment of all participants one and three years after inclusion into the study are described in former publications (Opaas and Hartmann, 2013; Opaas and Varvin, 2015; Opaas, Hartman, Wentzel-Larsen, & Varvin, 2016). 

Except for these three coordinated assessment points, all other assessment points were random, that is, not coordinated according to the time of inclusion into the study for each individual. Therefore, it is not meaningful to speak of missing data and sample size at each point in time. The statistical method chosen is suited for such data, where time between each measurement and the number of assessments varies among the participants. 

In Procedures, page 13, line number 249 to 252, an addition is made to make the nature of the data clearer.

---

## [Editor Report · Decision Letter 2]

16 Dec 2020

The 10-year course of mental health, quality of life, and exile life functioning in traumatized refugees from treatment start

PONE-D-20-12948R2

Dear Dr. Opaas,

We’re pleased to inform you that your manuscript has been judged scientifically suitable for publication and will be formally accepted for publication once it meets all outstanding technical requirements.

Kind regards,

James Curtis West, M.D.

Academic Editor

PLOS ONE

Additional Editor Comments (optional):

Thank you for your persistence in completing revisions to this manuscript.
---

## [Editor Report · Acceptance letter]

22 Dec 2020

PONE-D-20-12948R2 

The 10-year course of mental health, quality of life, and exile life functioning in traumatized refugees from treatment start 

Dear Dr. Opaas:

I'm pleased to inform you that your manuscript has been deemed suitable for publication in PLOS ONE. Congratulations! Your manuscript is now with our production department. 

Kind regards, 

on behalf of

Dr. James Curtis West 

Academic Editor

PLOS ONE